# A Traditional Korean Diet with a Low Dietary Inflammatory Index Increases Anti-Inflammatory IL-10 and Decreases Pro-Inflammatory NF-κB in a Small Dietary Intervention Study

**DOI:** 10.3390/nu12082468

**Published:** 2020-08-16

**Authors:** Phil-Kyung Shin, Seon-Joo Park, Myung Sunny Kim, Dae Young Kwon, Min Jung Kim, KyongChol Kim, Sukyung Chun, Hae-Jeung Lee, Sang-Woon Choi

**Affiliations:** 1CHA Bio Complex, CHA University, Seongnam 13488, Korea; pkshin34@gmail.com (P.-K.S.); rose6919@gmail.com (S.C.); 2Department of Food and Nutrition, College of BioNano Technology, Gachon University, Seongnam 13120, Korea; chris0825@hanmail.net; 3Research Group of Healthcare, Korea Food Research Institute, Wanju 55365, Korea; dykwon@kfri.re.kr (D.Y.K.); kmj@kfri.re.kr (M.J.K.); 4Department of Food Biotechnology, Korea University of Science and Technology, Daejeon 34113, Korea; 5GangNam Major Hospital, Seoul 06279, Korea; joyks71@gmail.com; 6Chaum Life Center, CHA University, Seoul 06062, Korea; 7Department of Nutrition, School of Public Health and Health Sciences, University of Massachusetts Amherst, Amherst, MA 01003, USA

**Keywords:** Korean diet, chronic inflammation, dietary inflammatory index, IL-10, NF-κB

## Abstract

Chronic low-grade inflammation may increase the risk of chronic disease, while diets rich in anti-inflammatory components may reduce it. To determine the anti-inflammatory properties of the traditional Korean diet (K-diet) that comprises high amounts of vegetables, fiber and phytochemicals, moderate amounts of legumes, and low amounts of animal fat, ten obese women aged 50–60 years were randomly assigned to the K-diet or control diet group. The control diet was a Westernized Korean diet commonly consumed in Korea, which is high in animal fat and protein. Subjects were housed in metabolic unit-like conditions during the 2-week intervention. Plasma was collected before and after the intervention to measure inflammatory cytokines using ELISA. The dietary inflammatory index (DII) was calculated based on nutrients and food intake. The DII score for the K-diet was lower than that of the control diet (−0.94 ± 1.39 vs. 1.04 ± 1.61, *p* < 0.001). In the K-diet group, anti-inflammatory interleukin (IL)-10 levels increased (4.45 ± 0.34 pg/mL vs. 5.94 ± 0.33 pg/mL, *p* = 0.0102), whereas pro-inflammatory nuclear factor kappa B (NF-κB) levels decreased (7.70 ± 0.62 pg/mL vs. 2.71 ± 0.49 pg/mL, *p* = 0.0015), but not in the control group. In the K-diet group, NF-κB levels negatively correlated with IL-10 levels (r = −0.794, *p* = 0.006). The K-diet has anti-inflammatory properties, and IL-10 and NF-κB are putative inflammatory markers for K-diet studies.

## 1. Introduction

Inflammation has been regarded as a part of the body’s defense mechanisms against common pathogens or injuries. However, low-grade chronic inflammation, which is commonly associated with obesity and insulin resistance [1,2], termed meta-inflammation [3], produces a steady subclinical inflammation throughout the body, ultimately resulting in major diseases such as cardiovascular disease, metabolic disorders and cancer [4]. This type of inflammation is characterized by a sustained presence of peripheral inflammatory markers such as C-reactive protein (CRP) and various inflammatory cytokines such as nuclear factor kappa B (NF-κB), interleukin (IL)-6, tumor necrosis factor α (TNFα) and interferon-γ (INF-γ). Various endogenous and exogenous factors including aging, gender, exercise and diet are known to influence this type of inflammation [5].

Fruits and vegetables as well as fiber, vitamin C, vitamin D and β-carotene have consistently shown to be associated with reducing chronic subclinical inflammation [6]. However, the effect of diet on inflammation is more complicated because various pro- and anti-inflammatory components may interact and modify inflammation [7]. The Western-style diet, which consists of high amounts of red meat, simple carbohydrates and refined grains, has been associated with higher levels of CRP and IL-6, whereas the Mediterranean diet, which consists of high amount of fresh fruits and vegetables, whole grains, nuts and fish along with high consumption of wine and olive oil, has been associated with lower levels of such pro-inflammatory markers [8]. Because the traditional Korean diet (K-diet) is high in vegetables and fibers, moderate to high in legumes and fishes, and low in red meat and calories [9], previous Korean nationwide studies have demonstrated the anti-oxidant and anti-inflammatory capability of K-diet with high phytochemical contents [10,11]. Even though a certain segment of Koreans still keep the traditional dietary pattern, the fast shift towards the Western style diet is occurring along with globalization in Korea [12].

Although the cause and effect relationship is unclear and the degree and duration of inflammation that can contribute to the development of diseases remain to be elucidated [4], emerging evidence indicates that maintaining an anti-inflammatory diet might be the most efficient strategy for reducing unhealthy inflammatory conditions [6]. As low-grade inflammation progresses continuously and ultimately damages essential organs, the long-term provision of anti-inflammatory components through diet could be the most suitable physiologic modality as well as a relatively inexpensive and easy way to incorporate into lifestyle.

To determine the effect of diet on inflammation, a dietary inflammatory index (DII) has been developed [8]. The original DII was developed to provide a tool that could categorize an individual’s diet into a spectrum from the highest to the lowest inflammatory diet. Thereafter, the DII has been further upgraded using literature that linked diet and inflammation as well as by incorporating cell culture, animal, and human studies [11,13]. The DII includes macro- and micro-nutrients as well as other bioactive food components such as flavonoids [10]. Thus, the DII has been reported to be associated with a wide range of health conditions including metabolic disorders, cardiovascular diseases, cancer and neurocognitive diseases [14].

Using a small dietary intervention study, we evaluated the anti-inflammatory properties of K-diet through the comparison of inflammatory biomarkers and DII with a current Westernized Korean diet [15].

## 2. Materials and Methods 

### 2.1. Subjects

Ten women aged 50–60 years were recruited and randomly and equally divided into two diet groups, control diet and K-diet [15]. The major selection criteria were the body mass index (BMI) between 25 and 30 kg/m^2^ and blood total cholesterol level between 200 and 270 mg/dL. The major exclusion criteria included individuals with diabetes mellitus, thyroid, or cardiovascular diseases as well as individuals under the lipid-lowering therapy or hormone replacement therapy. This human study was approved by CHA University Bundang Medical Center Institutional Review Board (BD 2015-107).

### 2.2. Methods

#### 2.2.1. Dietary Intervention

The provided K-diet was a traditional Korean diet previously defined by our group [9], while the control diet was a Westernized Korean diet, which is currently consumed in Korea. Diets were prepared using traditional cooking methods. For the 2-week dietary intervention, subjects were housed in a condition similar to the metabolic unit, where exercise and other activities were strictly controlled to make all conditions similar except diet. Diets were served as the traditional Korean table setting that consists of a rice bowel, a soup, and side dishes. Both diet recipes were such that the diets had same caloric value, 1800 kcal/day, which meets the estimated average requirement of Korean women aged 50–64 years old [16]. However, compared to the control diet, K-diet consisted of more vegetables, vegetable source protein and fat, and carbohydrate with lower glycemic index, and less fat, especially animal fat. Both groups were served three meals and two snacks per day. The K-diet group was provided with traditional drinks and soymilk, while the control group was provided coffee, juice and milk. Alcohol consumption was not allowed but a small amount of alcohol was used for cooking. The amount of consumed food was quantified by subtracting the amount of food leftover from the amount of food served. The amounts of consumed dietary nutrients and calories were calculated using the CAN-Pro 5.0 program (The Korean Nutrition Society, Seoul, Korea).

#### 2.2.2. Blood Chemistry

Complete blood count and blood chemistry profile were evaluated at the Department of Clinical Pathology at Kangnam CHA hospital using an automated hematology analyzer (Sysmex, Kobe, Japan) and an automated chemistry analyzer (Hitachi, Tokyo, Japan). CRP was measured with a latex agglutination test (Qualigent CRP, Sekisui Medical, Tokyo, Japan) and fibrinogen was measured with a turbidimetry test (ACL TOP Family system, Instrumentation Laboratory, Bedford, MA, USA).

#### 2.2.3. Plasma Cytokine Levels

Plasma levels of IL-6, IL-10, NF-κB, TNFα, and IFN-γ were measured using commercially available kits (IL-6 Human ELISA Kit, IL-10 Human ELISA Kit, NF kappaBp65 (Total) Human ELISA Kit, TNF alpha Human ELISA Kit, IFN gamma Human ELISA Kit, Invitrogen by Thermo (Fisher Scientific, Waltham, MA, USA) and the VersaMax Microplate Reader (Molecular Devices, San Jose, CA, USA).

#### 2.2.4. Calculation of the Dietary Inflammatory Index

DII was calculated using a published method [13,17], in which intakes of 45 food parameters (36 nutrients and 9 foods) were considered. For the present study, consumption of 32 food parameters (26 nutrients and 5 foods) (Table 1) per person per day was calculated using the CAN-Pro 5.0 program (The Korean Nutrition Society, Seoul, Korea) [17]. The remaining parameters were not included in the DII calculation due to insufficient food database or very low intake. Z-score ((Individual’s estimate of intake-standard global mean)/(standard deviation)) was calculated based on the mean and standard deviation of the 32 food parameters. Z-scores for each food parameter were converted into a percentile value and centered by doubling the value and subtracting ‘1’. The centered percentile value of each food parameter was multiplied by the inflammatory score of each parameter to achieve the ‘specific-food parameter DII score’. Finally, all of the ‘specific-food parameter DII scores’ were added to calculate the individual DII scores. A higher DII score represents a more inflammatory diet, while a lower DII score represents a less inflammatory diet.

### 2.3. Statistics

Descriptive statistics are presented as means ± standard errors. Data were analyzed using independent *t*-test to compare the differences between the two groups; paired *t*-test was used to determine the significant differences between values before and after dietary intervention in each group, and Spearman’s test was used to test correlation. Values were considered significant at *p* < 0.05. Data were analyzed using SPSS version 24 (IBM Corp., Armonk, NY, USA).

## 3. Results

### 3.1. Baseline Characteristics and Changes in Clinical Parameters after the Dietary Intervention

Before the dietary intervention, there were no significant differences in anthropometric measures and blood chemistry between two groups (Table 2) (modified from [15]). After the dietary intervention, total blood cholesterol and hemoglobin levels decreased in the K-diet group, while there was no change in these levels in the control group (Table 3) (modified from [15]).

### 3.2. Comparison of Nutrient Intake and Food Consumption between Two Diet Groups

Both groups consumed similar amount of calories but the K-diet group consumed more carbohydrate and protein and less animal fat compared to the control group. The K-diet group also consumed more fiber, vitamins and iron compared to the control group (Table 4) (modified from [15]). Regarding food consumption K-diet group consumed more total and whole grains, fruits and vegetables, legumes and tofu, nuts, and fish, but consumed less red meat, eggs and processed foods (Table 5) (modified from [15]).

### 3.3. Comparisons of DII Scores between Two Diet Groups

The DII score of the K-diet group was lower than that of the control group (1.04 ± 1.61 vs. −0.94 ± 1.39, *p* < 0.001) (Table 1). Among 26 nutrients and 5 foods, DII scores for the K-diet were lower than those for the control diet for 15 nutrients including fiber, B vitamins, selenium and zinc as well as two food items, garlic and green tea. In contrast DII scores for the control diet were lower than for K-diet for seven nutrients including carbohydrate, vitamin D and vitamin E as well as two food items, onion and pepper.

### 3.4. Changes in Inflammatory Markers and Their Associations with Other Parameters

At the baseline, levels of all measured inflammatory makers did not show differences except for NFκB levels, which were significantly higher in the K-diet group than in the control group (7.70 ± 0.62 pg/mL vs. 3.85 ± 0.95 pg/mL, *p* = 0.0092) (Figure 1). After the dietary intervention IL-10 levels increased (4.45 ± 0.34 pg/mL, vs. 5.94 ± 0.33 pg/mL, *p* = 0.0102) and NF-κB levels decreased (7.70 ± 0.62 pg/mL vs. 2.71 ± 0.49 pg/mL, *p* = 0.0015) in the K diet group but not in the control group (Figure 1). In the K-diet group, NF-κB levels were negatively correlated with IL-10 levels (r = −0.794, *p* = 0.006) but positively correlated with the total cholesterol levels (r = 0.636, *p* = 0.048) and fasting blood glucose levels (r = 0.648, *p* = 0.047). In contrast, there was no correlation between NF-κB and IL-10 levels in the control group. In the present study, CRP, fibrinogen, IL-6, TNFα, and INF-γ levels did not show significant changes in both diet groups (Table 6). Nevertheless, CRP levels showed a positive correlation with fasting insulin levels in both diet groups (K-diet group; r = 0.649 *p* = 0.048, control diet group: r = 0.632 *p* = 0.050).

## 4. Discussion

Accumulating evidence strongly suggests that diet plays a fundamental role in the modulation of chronic subclinical inflammation. The K-diet rich in anti-inflammatory components has been regarded as a healthy diet [10], which is expected to reduce chronic sustained inflammation.

In the present study, notably K-diet elevated the plasma IL-10 levels and lowered the NF-κB levels; further IL-10 levels were negatively correlated with the NF-κB levels in the K-diet group. IL-10, an immunoregulatory cytokine that has broad effects throughout the immune system, is the most important anti-inflammatory cytokine [18]. NF-κB is an essential mediator of inflammatory response and its activation in macrophages initiates the inflammatory cascade [19]. As IL-10 can block the activity of NF-κB, increased anti-inflammatory IL-10 due to K-diet consumption might down-regulate pro-inflammatory NF-κB.

Previous studies have demonstrated that IL-10 is an important protective cytokine against the chronic sustained inflammation commonly associated with aging, obesity, and diabetes mellitus. Evidence also suggests the cause and effect relationship between diet and IL-10. In an animal study, high-fat-diet-induced obesity significantly decreased IL-10 blood levels, while IL-10 treatment reduced inflammation induced by high-fat-diet [20]. In an IL-10-deficient mouse model, age-associated endothelial dysfunction occurs earlier with IL-10 deficiency, and IL-10 protected the age-associated increase in the expression of IL-6, a pro-inflammatory cytokine [21]. In a transgenic mouse model with muscle-specific IL-10 overexpression, IL-10 reduced insulin resistance and protected skeletal muscle from obesity-associated macrophage infiltration [22]. In murine models of autoimmunity, IL-10 deficiency demonstrated disease exacerbation, while IL-10 overexpression illustrated immunosuppressive properties [23]. Collectively, the elevation of IL-10 levels due to consumption of K-diet suggests that it may reduce inflammation and potentiate immunity. 

NF-κB is a key regulator of energy metabolism and associated inflammation. Studies using mouse models of obesity have shown that high-fat diets can increase NF-κB activation, resulting in chronic prolonged inflammation and the elevation of other circulating pro-inflammatory cytokine levels [24]. Recent evidence also indicates that the activation of NF-κB is critical for the development of insulin resistance [25]. Clinical studies have demonstrated that high-fat meals can induce NF-κB activation from peripheral blood mononuclear cells and obese individuals have higher levels of NF-κB compared to lean individuals [24]. Such studies have also provided evidence for a link between obesity and tumorigenesis conveyed through the NF-κB-dependent inflammatory response [24]. Thus, reduced plasma levels of NF-κB in the K-diet group may indicate an anti-inflammatory effect of K-diet consumption and its subsequent positive effects on obesity, insulin resistance or tumorigenesis. In the K-diet group total cholesterol levels were decreased in parallel with the decrease in NF-κB levels due to lower consumption of cholesterol and animal fat, which is similar to that reported in a previous study [26].

The average DII score of K-diet was −0.94 ± 1.39, which was lower than that of control group (1.04 ± 1.61). However, the DII score of control diet was still much lower than the reported score of maximally pro-inflammatory diet which is +7.98 [13]. The DII scores of nutrients and foods in the K-diet were lower when compared to those of the control diet for 15 nutrients and 2 foods. However, the DII scores of the control diet were also lower when compared to those of K-diet for 7 nutrients and 2 foods, suggesting that control diet was not a high pro-inflammatory diet. If we had designed an increased pro-inflammatory diet, we might have obtained higher differences in pro-inflammatory mediators such as CRP, fibrinogen, IL-6, TNF-α, and IFN-γ levels compared to the control group. As our IRB strongly suggested not providing an unhealthy diet as a control, we used a modest pro-inflammatory control diet, which resulted in minimally high DII score for the control diet.

Compared with the control diet group, subjects in the K-diet group consumed more quantities of iron (Table 4), which is known as a pro-inflammatory micronutrient [27]; the DII score for iron was significantly higher in the K-diet group than in the control group (Table 1). Nevertheless, hemoglobin levels decreased in the K-diet group but not in the control diet group. Although the K-diet group consumed more iron, its sources were plant based with low iron bioavailability (Table 4). Furthermore, individuals in the K-diet group consumed much less red meat than in the control group (Table 5). High iron consumption in the K-diet group increased the DII score but might not significantly contribute to inflammation due to low bioavailability of plant origin iron.

Carbohydrate DII score of the K-diet group was significantly higher than that of the control diet group (Table 1), because carbohydrate consumption was higher in the K-diet group than in the control diet group (Table 4). However, fasting blood glucose and insulin levels as well as HOMA-IR did not show significant changes in both groups, suggesting the effect of higher carbohydrate intake on inflammation might not be significant regardless of the higher DII score of carbohydrate. It is probably because the source of carbohydrate in the K-diet was mainly from the whole grains that have high dietary fiber content with low glycemic index and high anti-inflammatory properties [28] (Table 5).

Levels of TNF-α, IL6 and IFN-γ were not altered, which might be due to a limited subject number, a short duration, and modest pro-inflammatory properties of the control diet. TNF-α, a primary component of the obesity–diabetes mellitus link [1], is a major pro-inflammatory cytokine involved in the early inflammatory phase and induces NF-κB production [29]. Possibly, TNF-α levels did not change as IL-10 levels were elevated, which is a potent inhibitor of TNF-α [30]. TNF-α blockade is also known to reduce insulin resistance and improve lipid profiles in individuals with chronic sustained inflammation [31]. IL-6 also plays a central role in the pathogenesis of chronic diseases. The IL-6 signaling pathway mainly modulates the connection between aging and age-associated disorders [32]. In the present study, IL-6 levels did not change, probably because NF-κB, which is the main modulator of IL-6, was downregulated. IFN-γ is a master checkpoint regulator of inflammatory cytokine process [33] and orchestrates leukocyte attraction, natural killer (NK) cell activity and B cell functions [34]. In fact, IFN-γ activates the pro-inflammatory NF-κB pathway [35] and antagonized the anti-inflammatory IL-10 pathway [36]. Thus, up-regulated IL-10 and down-regulated NF-κB might result in unchanged IFN-γ levels.

CRP and fibrinogen levels also did not change. CRP is an acute-phase reactant protein produced mainly in the liver in response to inflammatory stimulation [37]. As CRP is influenced by diet, it has extensively been used as a major inflammatory biomarker that determines the dietary effects on inflammation. The association between CRP and diet mainly depends on the balance between pro-inflammatory and anti-inflammatory components of diet [37]. In the present study, CRP levels did not change significantly, probably because the CRP levels of all subjects were within normal range from the beginning and both K-diet and control diet did not have high DII scores. Even though obesity is known to be associated with the elevation of CRP, our subjects whose BMIs were between 25 and 30 kg/m^2^ (Asian obesity criteria is BMI 25 and higher) showed normal CRP levels. One positive finding was that CRP levels had a positive correlation with fasting blood insulin levels in both diet groups, which is similar to the previous report regarding the chronic systemic inflammation, BMI, insulin resistance and CRP [38]. Fibrinogen, an independent risk factor for cardiovascular disease, is also a positive acute-phase reactant synthesized in the liver. Fibrinogen synthesis is triggered by the ingestion of a meal [39]. Westernized diet increases [40] and plant-based diet reduces serum fibrinogen levels [41]. However, in the present study there was no significant change in the fibrinogen levels. Further study is necessary to determine the anti-inflammatory effects of K-diet on cardiovascular disease.

This study aimed to determine the anti-inflammatory features of K-diet by measuring the levels of blood inflammatory mediators and DII scores. Owing to the limited subject number and moderately pro-inflammatory control diet, significant results obtained were limited. However, levels of at least two major inflammatory cytokines showed significant differences along with significantly different DII scores. Future investigations with an increased number of subjects are warranted to clarify the anti-inflammatory effects of K-diet.

## Figures and Tables

**Figure 1 nutrients-12-02468-f001:**
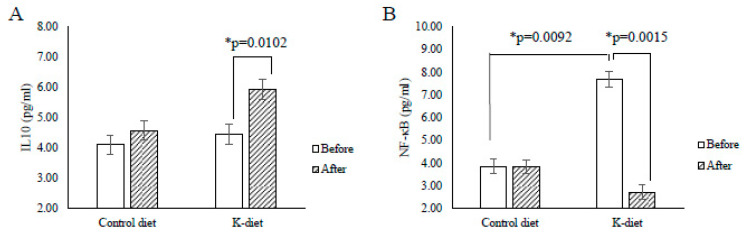
Changes in cytokine levels due to K-diet consumption. (**A**) Interleukin (IL)-10 levels significantly increased in the K-diet group, but not in the control group after the 2-week dietary intervention. (**B**) Before the intervention, the average nuclear factor kappa B (NF-κB) levels in the K-diet group were significantly higher when compared to that in the control diet group. After the dietary intervention, NF-κB levels were significantly reduced only in the K-diet group. All values are given as mean ± SE (*n* = 5). * *p* < 0.05 by paired *t*-test.

**Table 1 nutrients-12-02468-t001:** Comparison of dietary inflammatory index (DII) between control and Korean diet (K-diet).

Food Parameters	Control Diet	K-Diet	*p*-Value
	Mean ± SD	Mean ± SD	
Energy (kcal)	0.01 ± 0.12	−0.01 ± 0.09	0.4713
Carbohydrate (g)	−0.04 ± 0.05	0.03 ± 0.04	<0.0001
Protein (g)	0.00 ± 0.01	0.00 ± 0.01	0.0123
Total fat (g)	−0.06 ± 0.17	0.05 ± 0.16	0.0003
Cholesterol (mg)	0.05 ± 0.09	−0.04 ± 0.11	<0.0001
Saturated fat (g)	−0.05 ± 0.23	0.05 ± 0.20	0.0093
MUFA (g)	0.00 ± 0.01	0.00 ± 0.01	0.0041
PUFA (g)	0.07 ± 0.18	−0.06 ± 0.19	<0.0001
n-3 Fatty acids (g)	0.04 ± 0.25	−0.04 ± 0.25	0.0764
n-6 Fatty acids (g)	0.02 ± 0.09	−0.02 ± 0.09	0.0166
Fiber (g)	0.32 ± 0.23	−0.31 ± 0.21	<0.0001
Vitamin A (RE)	0.04 ± 0.21	−0.04 ± 0.24	0.0471
β-Carotene (μg)	0.11 ± 0.31	−0.10 ± 0.34	0.0003
Thiamin (mg)	0.04 ± 0.05	−0.04 ± 0.04	<0.0001
Riboflavin (mg)	0.02 ± 0.03	−0.02 ± 0.04	<0.0001
Niacin (mg)	0.12 ± 0.08	−0.12 ± 0.08	<0.0001
Vitamin B6 (mg)	0.09 ± 0.24	−0.08 ± 0.13	<0.0001
Folic acid (μg)	0.08 ± 0.08	−0.08 ± 0.08	<0.0001
Vitamin B12 (μg)	−0.03 ± 0.07	0.02 ± 0.05	<0.0001
Vitamin C (mg)	0.03 ± 0.28	−0.02 ± 0.20	0.2567
Vitamin D (μg)	−0.11 ± 0.25	0.12 ± 0.21	<0.0001
Vitamin E (mg)	−0.11 ± 0.26	0.11 ± 0.16	<0.0001
Mg (mg)	0.03 ± 0.24	−0.03 ± 0.31	0.2338
Fe (mg)	−0.01 ± 0.01	0.01 ± 0.01	<0.0001
Se (μg)	0.07 ± 0.18	−0.07 ± 0.15	<0.0001
Zn (mg)	0.03 ± 0.06	−0.04 ± 0.04	<0.0001
Garlic (g)	0.15 ± 0.22	−0.14 ± 0.16	<0.0001
Onion (g)	−0.09 ± 0.15	0.10 ± 0.14	<0.0001
Ginger (g)	0.01 ± 0.33	0.00 ± 0.17	0.7513
Green/black tea (g)	0.27 ± 0.16	−0.26 ± 0.16	<0.0001
Pepper (g)	−0.06 ± 0.04	0.06 ± 0.05	<0.0001
Alcohol(g)	0.00 ± 0.15	0.00 ± 0.17	0.9504
DII	1.04 ± 1.61	−0.94 ± 1.39	<0.0001

RE: retinol equivalents; MUFA, monounsaturated fatty acids; PUFA Polyunsaturated fatty acid, DII: dietary inflammatory index; data are expressed as means ± SD (standard deviation). All *p*-values represent differences between control group and Korean food group.

**Table 2 nutrients-12-02468-t002:** Subject profiles.

Clinical Parameters	Control Diet (*n* = 5)	K-Diet (*n* = 5) *
Age	54.60 ± 0.87	52.8 ± 1.02
Weight (kg)	66.48 ± 2.02	64.36 ± 2.12
BMI (Kg/m^2^)	27.30 ± 0.93	26.00 ± 0.64
Waist circumference (cm)	90.60 ± 1.54	85.40 ± 3.64
Total cholesterol (mg/dL)	209.40 ± 4.74	239.40 ± 15.14
LDL-cholesterol (mg/dL)	142.20 ± 5.40	138.40 ± 24.91
HDL-cholesterol (mg/dL)	46.56 ± 4.05	57.65 ± 3.40
Triglyceride (mg/dL)	103.20 ± 12.99	237.80 ± 116.24
Fasting blood glucose (mg/dL)	92.40 ± 4.15	93.40 ± 3.31
Insulin (mU/L)	6.64 ± 1.05	10.46 ± 2.09
CRP (mg/dL)	0.12 ± 0.03	0.12 ± 0.04
Fibrinogen (mg/dL)	292.40 ± 27.18	283.20 ± 20.73

* All *p*-values between two groups were not significant.

**Table 3 nutrients-12-02468-t003:** Comparison of clinical endpoints before and after the intervention.

	Control Diet			K-Diet		
Variable	Baseline	2 Weeks	*p* Value	Baseline	2 Weeks	*p* Value
BMI (kg/m^2^)	27.30 ± 0.93	26.97 ± 0.84	NS	26.00 ± 0.64	25.62 ± 0.53	NS
Weight (Kg)	66.48 ± 2.02	65.68 ± 1.97	NS	64.36 ± 2.12	63.36 ± 1.82	NS
Waist circumference (cm)	90.60 ± 1.54	88.20 ± 0.85	NS	85.40 ± 3.64	83.80 ± 2.05	NS
Total cholesterol (mg/dL)	209.40 ± 4.74	229.80 ± 8.12	NS	239.40 ± 15.14	198.20 ± 13.25	0.0163
LDL cholesterol (mg/dL)	142.20 ± 5.40	146.60 ± 7.09	NS	138.40 ± 24.91	123.60 ± 13.05	NS
HD -cholesterol (mg/dL)	46.56 ± 4.05	55.12 ± 5.53	NS	57.65 ± 3.40	49.52 ± 9.03	NS
Triglyceride (mg/dL)	103.20 ± 12.99	140.40 ± 16.10	NS	237.80 ± 116.24	125.40 ± 16.36	NS
Fasting glucose (mg/dL)	92.40 ± 4.15	89.60 ± 3.39	NS	93.40 ± 3.31	82.20 ± 3.92	NS
Insulin (mU/L)	6.64 ± 1.05	9.00 ± 1.20	NS	10.46 ± 2.09	7.40 ± 1.34	NS
HOMA-IR	1.48 ± 0.18	1.98 ± 0.24	NS	2.47 ± 0.55	1.46 ± 0.20	NS
WBC (×10^3^/μL)	5.97 ± 0.69	5.70 ± 0.57	NS	6.43 ± 0.57	5.42 ± 0.68	NS
Hb (g/dL)	13.52 ± 0.18	13.34 ± 0.34	NS	13.38 ± 0.34	12.86 ± 0.49	0.0348

NS: not significant.

**Table 4 nutrients-12-02468-t004:** Comparison of macro and micro-nutrient intake between control and K-diet group.

Nutrients	Control Diet (*n* = 5)	K-Diet (*n* = 5)	*p* Value
Energy (kcal)	1775.5 ± 25.5	1740.2 ± 12.7	NS
Carbohydrate (% of energy)	57 ± 0.6	63.7 ± 0.4	<0.0001
Fiber (g)	27.1 ± 1.3	41.5 ± 0.4	<0.0001
Protein (% of energy)	15.7 ± 0.2	17.1 ± 0.3	<0.0001
Animal based protein (% of energy)	7.3 ± 0.3	4.9 ± 0.3	<0.0001
Plant based protein	8.4 ± 0.1	12.2 ± 0.1	<0.0001
Fat (% of energy)	27.4 ± 0.4	19.2 ± 0.3	<0.0001
Animal based fat (% of energy)	10.4 ± 0.5	2.3 ± 0.2	<0.0001
Plant based fat (% of energy)	17.1 ± 0.3	16.9 ± 0.3	NS
Cholesterol (mg)	447.3 ± 30	182.9 ± 11	<0.0001
Vitamins			
Vitamin A (µg RE)	1246.4 ± 42.4	1497.7 ± 71.4	0.0031
Vitamin C (mg)	177.4 ± 6.6	180.1 ± 3.8	NS
Thiamin (mg)	1 3 ±0	1.7 ± 0	<0.0001
Riboflavin (mg)	1.3 ± 0	1.5 ± 0	<0.0001
Niacin (mg)	16.4 ± 0.3	24.9 ± 0.4	<0.0001
Iron (mg)	15.3 ± 0.3	22.6 ± 0.7	<0.0001
Animal based iron (mg)	3.6 ± 0.2	4 ± 0.7	NS
Plant based iron (mg)	11.7 ± 0.2	18.6 ± 0.4	<0.0001

NS: not significant.

**Table 5 nutrients-12-02468-t005:** Comparison of food consumption between control and K-diet group.

Food	Control Diet (*n* = 5)	K-Diet (*n* = 5)	*p* Value
Total grains	217.4 ± 5.1	277.7 ± 3.9	<0.0001
Whole grains	0.4 ± 0.1	267.9 ± 4.9	<0.0001
Fruits and vegetables	405.1 ± 7	543.2 ± 10.3	<.0001
Legumes and tofu	40 ± 4.6	63.4 ± 4.6	0.0004
Nuts	2.6 ± 0.6	21.4 ± 3.8	<0.0001
Seaweeds	15.7 ± 2.8	24.5 ± 3.9	0.0708
Fishes and shell	35.4 ± 3.9	53.2 ± 5.3	0.0073
Meats	57.3 ± 4.6	10 ± 2.2	<0.0001
Red meats	48.8 ± 4.2	5.4 ± 1.3	<0.0001
Processed foods	21.9 ± 3.4	0 ± 0	<0.0001

**Table 6 nutrients-12-02468-t006:** Comparison of inflammatory markers before and after the intervention.

	Control Diet (*n* = 5)			K-Diet (*n* = 5)		
Inflammation Markers	Baseline	2 Weeks	*p* Value	Baseline	2 Weeks	*p* Value
CRP (mg/dL)	0.12 ± 0.03	0.11 ± 0.03	NS	0.12 ± 0.04	0.07 ± 0.02	NS
Fibrinogen (mg/dL)	292.4 ± 27.18	283.4 ± 25.76	NS	283.2 ± 20.73	253.8 ± 17.72	NS
TNF-α (pg/mL)	2.21 ± 0.32	1.93 ± 0.12	NS	1.80 ± 0.24	2.27 ± 0.24	NS
NFκB (pg/mL)	3.85 ± 0.95	3.84 ± 0.61	NS	7.70 ± 0.62 *	2.71 ± 0.49	0.0015
IL-6 (pg/mL)	0.68 ± 0.12	0.50 ± 0.22	NS	0.78 ± 0.06	1.34 ± 0.89	NS
IL-10 (pg/mL)	4.10 ± 0.31	4.58 ± 0.30	NS	4.45 ± 0.34	5.94 ± 0.33	0.0102
IFN-γ (pg/mL)	4.51 ± 1.03	2.65 ± 0.66	NS	2.74 ± 0.28	2.35 ± 0.21	NS

* *p* = 0.0092 compared to the baseline of control diet group. NS: not significant.

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
