# Peer review of "A Traditional Korean Diet with a Low Dietary Inflammatory Index Increases Anti-Inflammatory IL-10 and Decreases Pro-Inflammatory NF-κB in a Small Dietary Intervention Study"

_nutrients, 2020, doi:10.3390/nu12082468_

Round 1

Reviewer 1 Report

My main concern with this paper (and it's not really 'major') is that the title reflects a larger and more comprehensive study, where it would be more honestly reflected as a 'pilot' or 'preliminary' study because of the small number of subjects (10) involved. I suggest the title be changed to at least include a subtitle to reflect this.

Secondly, I am a bit disturbed that the introduction does not reflect some of the earlier studies that defined this area, such as by those by Hotamisligil and colleagues in 1994 and the early 2000s. This could have set the stage for the use of the term 'metaflammation' when introducing the concept of low grade, systemic inflammation, bringing the study more into focus.

Third, there is no recognition of a number of studies published in 'Obesity Reviews' in the 2000s, which outlined the shift to this area of study by listing pro and anti-inflammatory foods from the literature. This is reflected in the fact that obesity was the first area considered as being the 'cause' of metaflammation (by Hotamisligil) in 1994

Lastly (and in recognition of the fact that English is probably a second language for the authors), there are many occasions in the article where definite words ('a', 'the', 'and'  'been'etc.) are omitted or misused. (eg. line 39,40 in abstract; 51, 56, p2; 102ps; 'markers' line 165 p3. line 184, 217, 218, 220 p 5 etc.

Author Response

Answers to critiques from the reviewer 1

Critique 1) My main concern with this paper (and it's not really 'major') is that the title reflects a larger and more comprehensive study, where it would be more honestly reflected as a 'pilot' or 'preliminary' study because of the small number of subjects (10) involved. I suggest the title be changed to at least include a subtitle to reflect this.

Answer 1)

As suggested the title has been changed to “A traditional Korean diet with a low dietary inflammatory index increases circulating anti-inflammatory IL-10 and decreases pro-inflammatory NF-κB in a small dietary intervention study.”

Critique 2) Secondly, I am a bit disturbed that the introduction does not reflect some of the earlier studies that defined this area, such as by those by Hotamisligil and colleagues in 1994 and the early 2000s. This could have set the stage for the use of the term 'metaflammation' when introducing the concept of low grade, systemic inflammation, bringing the study more into focus.

Answer 2)

Thank you for suggesting meta-inflammation. It has been added to the introduction section (Line 46-47) and references were cited.

“low-grade chronic inflammation, which is commonly associated with obesity and insulin resistance [1,2], termed meta-inflammation [3], produces a steady subclinical inflammation throughout the body”

Critique 3) Third, there is no recognition of a number of studies published in 'Obesity Reviews' in the 2000s, which outlined the shift to this area of study by listing pro and anti-inflammatory foods from the literature. This is reflected in the fact that obesity was the first area considered as being the 'cause' of metaflammation (by Hotamisligil) in 1994

Answer 3)

Sentences regarding meta-inflammation and obesity-diabetes link was added to the introduction section (Line 46-47) and discussion section (Line 259-260).

Critique 4) Lastly (and in recognition of the fact that English is probably a second language for the authors), there are many occasions in the article where definite words ('a', 'the', 'and'  'been'etc.) are omitted or misused. (eg. line 39,40 in abstract; 51, 56, p2; 102ps; 'markers' line 165 p3. line 184, 217, 218, 220 p 5 etc.

Answer 4) Corrected

Reviewer 2 Report

The movement of examinations of the DII from epidemiology studies to intervention trials is a positive next step taken by this research. There are concerns however both in the study design and presentation of findings.

Abstract:
Line 27: insert "the" between determine and anti-inflammatory

Line 29: "caloric" vs "calorific"

The Korean diet is described but not the Westernized diet - would leave descriptions out of Abstract or include for both.

Line 32: insert "the" between after and intervention

Line 32: Insert "The" to start sentence

Line 33: Consumed food parameters? What does that mean - plate waste, menus?

Line 35: do not need "in plasma...K-diet group"

Lines 36-38: Do you need the correlation of biomarkers given you already say they significantly increased or decreased? Is cholesterol important given focus on inflammation?
This just seems like an awkward sentence/placement of sentence.

Line 39: Since the diet was a known diet, given intervention set-up, would put DII score first and then plasma findings. Also you should put the lowest value first in the parentheses since you introduce them in the text as K-diet lower than control diet.

Introduction:

I believe the DII has been registered - the authors should check with the creators of the DII to see if a registration mark is needed.

Line 81: need references for this sentence.

Line 82: You introduce K-diet as an abbreviation in the abstract but not in the Introduction. You need to introduce the abbreviation on the first use of the label "Korean diet" in the body of the manuscript.

Given reference, it seems like this analysis is part of a larger study and that needs to be noted in the Introduction too. What was the larger study, its purpose, etc.?

Line 82-83: This is an inadequate purpose statement - you didn't just examine the Korean diet, you examined it against a Western Korean diet, you did so in a controlled environment and measured specific biomarkers. You need to be more detailed and specific about your research design and outcomes of interest in this statement.

Given your examination of the K-diet vs Western Korean diet, your Introduction needs an overview of this apparent shift that is happening/multiple Korean diets that exist/are emerging to show there is a need to look at the inflammatory potential of the K-diet vs a Western Korean diet.

Methods

Did you complete a power calculation to show this small of a sample size was adequate to detect differences in the outcomes of interest? Without a power calculation the small sample size is a serious study design concern.

Line 96-97: But your introduction (info on national dietary intake) makes it seem like everyone is eating a healthy diet. You need to clarify this shift in dietary intake in the introduction (see comment there).

How many meals/snacks were served - were these the same between the two groups? Were traditional cooking methods used with both groups?

Did researchers/nutrition professionals do the cooking and plate waste calculations?

What was the total calories for each group? Is this in line with caloric intake in the population?

The menu from each group as a supplemental table would be helpful (see comments in Results regarding need for description of diets to be in Methods and description of what what actually eaten vs. provided in Results).

Biochemistry: Were any efforts made to make sure short-term inflammation did not influence biomarkers (colds, virus, taking NSAIDS)?

Results:

Lines 159-164: This is less results and more methods given these were prescribed diets. Methods should indicate how the diets differed and then results can indicate how much was eaten and how that impacted control vs intervention intake (e.g., both ate 80% of meals or arms differed in amount consumed, were there any patterns in what was and was not eaten within arms and across arms, impact on nutrient intake).

Lines 170-175:  gain this is less results and more methods since the menus were developed by the researchers.

Discussion:

What amount of an increase in IL-10 is needed for clinical significance? You talk about increases, but how much is needed to truly impact chronic inflammation/chronic disease? Same with other biomarker that showed statistically significant change - but is it of clinical significance?

Lines 230-232: This sentence doesn't make sense. At baseline, levels were higher in K-diet vs control and there was no significant difference in pre-post control levels so why the need for this explanation/what are you trying to explain?

Were hemoglobin levels reduced in the K-diet enough to be clinically worrisome (how many of the 40-50 yo population were premenopausal, what if this were a younger population)?

When discussing carbohydrate intake, fiber should also be discussed, especially as you suggest the source of carbohydrates in the K-diet are whole grains with low glycemic load.

Tables:
Table 1 - can you indicate p-values with a footnote vs a column of all NS?

Table 6 - need to bold and star IL-10 under K-diet

Author Response

Answers to Critiques from the Reviewer 2

The movement of examinations of the DII from epidemiology studies to intervention trials is a positive next step taken by this research. There are concerns however both in the study design and presentation of findings.

Abstract:

Critique 1) Line 27: insert "the" between determine and anti-inflammatory

Answer 1) inserted (Line 28)

Critique 2) Line 29: "caloric" vs "calorific"

Answer 2) this sentence has been removed due to words limit

Critique 3) The Korean diet is described but not the Westernized diet - would leave descriptions out of Abstract or include for both.

Answer 3) the description for the Westernized Korean diet has been added to the Line 31-32.   

The control diet was a Westernized Korean diet commonly consumed in Korea, which is high in animal fat and protein.

Critique 4) Line 32: insert "the" between after and intervention

Answer 4) inserted (Line 34)

Critique 5) Line 32: Insert "The" to start sentence

Answer 5) inserted (Line 34) 

Critique 6) Line 33: Consumed food parameters? What does that mean - plate waste, menus?

Answer 6) This sentence has been changed (Line 35).

The dietary inflammatory index (DII) was calculated based on nutrients and foods intake

Please refer to “Shin D, Lee KW, Brann L, Shivappa N, Hébert JR. Dietary inflammatory index is positively associated with serum high-sensitivity C-reactive protein in a Korean adult population. Nutrition. 2019;63-64:155-161. doi:10.1016/j.nut.2018.11.016”  

Critique 7) Line 35: do not need "in plasma...K-diet group"

Answer 7) removed

Critique 8) Lines 36-38: Do you need the correlation of biomarkers given you already say they significantly increased or decreased? Is cholesterol important given focus on inflammation?

This just seems like an awkward sentence/placement of sentence.

Answer 8) the sentence has been removed

Critique 9) Line 39: Since the diet was a known diet, given intervention set-up, would put DII score first and then plasma findings. Also you should put the lowest value first in the parentheses since you introduce them in the text as K-diet lower than control diet.

Answer 9) has been changed as suggested (Line 36)

(-0.94±1.39 vs 1.04±1.61, vs p<0.001)

Introduction:

Critique 10) I believe the DII has been registered - the authors should check with the creators of the DII to see if a registration mark is needed.

Answer 10) To the best of our knowledge, DII has been developed by Shivappa et al (2014). It is not patented/registered and can be used freely. A large number of studies are using DII without any registration. (Veronese et al. 2017; Vahid et al. 2017; Shin et al. 2018)

References

  • Shivappa, N., Steck, S., Hurley, T., Hussey, J., & Hébert, J. (2014). Designing and developing a literature-derived, population-based dietary inflammatory index. Public Health Nutrition, 17(8), 1689-1696. doi:10.1017/S1368980013002115
  • Veronese N, Stubbs B, Koyanagi A, Hebert JR, Cooper C, Caruso MG, et al. Pro-inflammatory dietary pattern is associated with fractures in women: an eight-year longitudinal cohort study. Osteoporos Int. 2017.
  • Vahid F, Shivappa N, Hekmatdoost A, Hebert JR, Davoodi SH, Sadeghi M. Association between maternal dietary inflammatory index (DII) and abortion in Iranian women and validation of DII with serum concentration of inflammatory factors: case-control study. Appl Physiol Nutr Metab. 2017;42(5):511–6.
  • Shin D, Lee KW, Brann L, Shivappa N, Hébert JR. Dietary inflammatory index is positively associated with serum high-sensitivity C-reactive protein in a Korean adult population. Nutrition. 2019;63-64:155-161. doi:10.1016/j.nut.2018.11.016

Critique 11) Line 81: need references for this sentence.

Answer 11) a reference has been added to the Line 82.

Hebert, J.R.; Shivappa, N.; Wirth, M.D.; Hussey, J.R.; Hurley, T.G. Perspective: The Dietary Inflammatory Index (DII)-Lessons Learned, Improvements Made, and Future Directions. Adv Nutr 2019, 10, 185-195, doi:10.1093/advances/nmy071

Critique 12) Line 82: You introduce K-diet as an abbreviation in the abstract but not in the Introduction. You need to introduce the abbreviation on the first use of the label "Korean diet" in the body of the manuscript.

Answer 12) done as suggested (Line 62-63)

Because the traditional Korean diet (K-diet) is high in vegetables and fibers,  

Critique 13) Given reference, it seems like this analysis is part of a larger study and that needs to be noted in the Introduction too. What was the larger study, its purpose, etc.?

Answer 13) It was an independent pilot study which was mainly focused on determining the epigenetic effects of K-diet. In fact, a larger study was performed with a different study design thereafter but we cannot include it to this manuscript because they have somewhat different aims and the data analysis is still underway.

Critique 14) Line 82-83: This is an inadequate purpose statement - you didn't just examine the Korean diet, you examined it against a Western Korean diet, you did so in a controlled environment and measured specific biomarkers. You need to be more detailed and specific about your research design and outcomes of interest in this statement.

Answer 14) We summarized our study at the end of introduction section as suggested (Line 83-85)

Critique 15) Given your examination of the K-diet vs Western Korean diet, your Introduction needs an overview of this apparent shift that is happening/multiple Korean diets that exist/are emerging to show there is a need to look at the inflammatory potential of the K-diet vs a Western Korean diet.

Answer 15) added to the Line 62 to 67

Because the traditional Korean diet (K-diet) is high in vegetables and fibers, moderate to high in legumes and fishes, and low in red meat and calorie [9], previous Korean nationwide studies have demonstrated the anti-oxidant and anti-inflammatory capability of K-diet with high phytochemical contents [10,11]. Even though a certain segment of Koreans still keep the traditional dietary pattern, the fast shift towards the Western style diet is going on along with the globalization in Korea [12].

Methods

Critique 16) Did you complete a power calculation to show this small of a sample size was adequate to detect differences in the outcomes of interest? Without a power calculation the small sample size is a serious study design concern.

Answer 16) Unfortunately no. We did not do a power calculation for the inflammatory markers because the main purpose of this study was to determine the genome-wide epigenetic changes by the K-diet. Five subjects per group was OK to determine the changes. Nevertheless, we got the significant difference with our subjects

We rented a mansion building at the foot of the mountain for the intervention study. Although we have a small sample size, lifestyle and diet were controlled in detail through a 24-hour monitoring system every day to live with research managers. In large studies, it is difficult to control those things.

The number of study residents during the study period was at least 1 nurse, 1or 2 nutritionist, 2 cookers, and 2 study managers. The study was conducted in three laboratories and many researchers participated as coordinators and advocates. Please see our method section. All subjects were housed in a condition similar to the metabolic unit, where exercise and other activities were strictly controlled to make all conditions similar except diet. Diet and snack were prepared by dietitians. We weighed all foods and served and measured the leftover.

Critique 17) Line 96-97: But your introduction (info on national dietary intake) makes it seem like everyone is eating a healthy diet. You need to clarify this shift in dietary intake in the introduction (see comment there).

Answer 17) The dietary shift has been shown in the introduction Line (Line 65-67).

Critique 18) How many meals/snacks were served - were these the same between the two groups? Were traditional cooking methods used with both groups?

Answer 18) Line 107: Both groups were served three meals and two snacks per day.

“Both groups were served 3 meals and 2 snacks per day. Two groups were given different type of meals and snacks. In table 2, we have shown the difference of food intake between two groups. All of the food provided to the k-food group were prepared using a traditional cooking method and few menus of the control groups were prepared with a traditional cooking method (boiling, less grease, seasoned vegetables) because it was difficult to provide only western style diet or fusion food to middle aged women who are more familiar to Korean dishes”

Critique 19) Did researchers/nutrition professionals do the cooking and plate waste calculations?

Answer 19) Yes. All cooking, calculation of serving amount, and amount of leftover food were done by nutritionists and nutrition professionals.

Critique 20)  What was the total calories for each group? Is this in line with caloric intake in the population?

Answer 20) Line 104-105: 1800 kcal/day, which meets the estimated average requirement of Korean women aged 50-64 years old [16].

We provided 1800kcal/day which meets the estimated average requirements (EAR) of Korean women aged 50-64 years old. It is similar to the average intake of Koreans around that age. According to KNHANES 2013-2015 data, the average energy intake of 50-69 years Korean women was 1,733.3 ± 14.4 kcal.  

Please refer to

Yun S, Kim HJ, Oh K.   Trends in energy intake among Korean adults, 1998-2015: Results from the Korea National Health and Nutrition Examination Survey.  Nutr Res Pract. 2017 Apr;11(2):147-154. https://doi.org/10.4162/nrp.2017.11.2.147

Critiques 21) The menu from each group as a supplemental table would be helpful (see comments in Results regarding need for description of diets to be in Methods and description of what what actually eaten vs. provided in Results).

Answer 21) Unfortunately, there are more than 100 menus for each group, so it is very difficult for us to show the entire menu as a table in our manuscript. Also, the menus for k-diet group is under patent pending so we cannot open that information.

Critique 22) Biochemistry: Were any efforts made to make sure short-term inflammation did not influence biomarkers (colds, virus, taking NSAIDS)?

Answer 22) Nurses were with subjects for two weeks and doctors were standby for two weeks to cope with any unexpected event or medical emergency but nothing has happened, neither extra-medication nor common viral infection.

Results:

Critique 23) Lines 159-164: This is less results and more methods given these were prescribed diets. Methods should indicate how the diets differed and then results can indicate how much was eaten and how that impacted control vs intervention intake (e.g., both ate 80% of meals or arms differed in amount consumed, were there any patterns in what was and was not eaten within arms and across arms, impact on nutrient intake).

Answer 23) Data from the provided nutrients can be a method but data from the consumed nutrients can be a result.  Table 3 is the comparison of consumption between the two groups. Thus, this table is a result after their consumption whatever we may have provided.

In fact, we asked the participants to finish the provided amount of food. Most of the subjects consumed everything, and although there were few cases of leftovers due to personal dislike of certain dishes, we weighed their leftovers and calculated the exact amount of their food intake. The difference of the food intake was not due to the leftovers, but it was because of the difference of the menu provided to each group. However, we do want to emphasize that almost 100% of the people ate everything we gave them.

Critique 24) Lines 170-175:  gain this is less results and more methods since the menus were developed by the researchers.

Answer 24) Even though the menus were developed by us, we calculated the DII after the experiment based on their consumption. Thus it could be better to be in the result.

Discussion:

Critique 25) What amount of an increase in IL-10 is needed for clinical significance? You talk about increases, but how much is needed to truly impact chronic inflammation/chronic disease? Same with other biomarker that showed statistically significant change - but is it of clinical significance?

Answer 25) In fact we do not know the clinically significant changes in IL-10 levels. In this study we just evaluated the anti-inflammatory effect by comparing with control. IL-6 and IL-10 levels are available at the clinic but the clinical significance is not clearly established yet, especially on their value changes.

Critique 26) Lines 230-232: This sentence doesn't make sense. At baseline, levels were higher in K-diet vs control and there was no significant difference in pre-post control levels so why the need for this explanation/what are you trying to explain?

Answer 26) the sentence has been removed.

Critique 27) Were hemoglobin levels reduced in the K-diet enough to be clinically worrisome (how many of the 40-50 yo population were premenopausal, what if this were a younger population)?

Answer 27) The hemoglobin level was still within normal range after the intervention (Table 2). In the present study subjects were more than 50 years old.  In the other companion study whose subjects were 30-50 years old women, the K-diet did not induce anemia.    

Critique 28) When discussing carbohydrate intake, fiber should also be discussed, especially as you suggest the source of carbohydrates in the K-diet are whole grains with low glycemic load.

Answer 28) added to the Line 256-257 with a reference

“mainly from the whole grains that has high dietary fiber content with low glycemic index and high anti-inflammatory property [28] (Table 4).”

Tables:

Critique 29) Table 1 - can you indicate p-values with a footnote vs a column of all NS?

Answer 29) done as suggested

Critique 30) Table 6 - need to bold and star IL-10 under K-diet

Answer 30) done as suggested